# Utilizing NU*track* to Access the Activity Levels in Pigs with Varying Degrees of Genetic Potential for Growth and Feed Intake

**DOI:** 10.3390/ani13101581

**Published:** 2023-05-09

**Authors:** Dalton Obermier, Melanie Trenahile-Grannemann, Ty Schmidt, Tom Rathje, Benny Mote

**Affiliations:** 1Department of Animal Science, University of Nebraska-Lincoln, Lincoln, NE 68588, USA; 2DNA Swine Genetics, 2415 13th Street, Columbus, NE 68601, USA

**Keywords:** activity, cost, efficiency, NU*track*, pigs

## Abstract

**Simple Summary:**

Precision livestock farming has been shown to not only be beneficial to producer profitability and production efficiency, but also in improving social acceptance and sustainability in livestock production. The research illustrated here utilizes an advanced computer vision system (NU*track*) that allows for the individual identification and tracking of pigs in a standard commercial setting. Traits that would otherwise be impractical to measure and therefore absent from selection criteria, such as the activity level, are easily obtained with this system. The data here suggest that progeny activity levels are impacted by sire EBVs for production traits. In the current study, pigs from boars with superior growth and feed intake EBVs were less active and faster growing compared to all other groups. Further work may lead to genetic selection for activity traits to decrease RFI levels and ultimately improve production rates through reduced calorie expenditure. Activity differences observed across sexes may contribute to managerial improvements in feeding barrows vs. gilts. Ultimately, furthering the objective tracking of activity patterns with NU*track* may help producers overcome difficulties that prevent them from maximizing feed efficiency; with potential for additional benefits to both the producer and the animal.

**Abstract:**

Feed cost accounts for over two-thirds of the variable cost of production. In order to reduce feed costs without sacrificing production numbers, feed efficiency must be improved. Calorie expenditure has been difficult to quantify in the past but is understood to impact residual feed intake (RFI) greatly. The objective of this work was to utilize an advanced computer vision system to evaluate activity levels across sex and sire groups with different expected breeding value combinations for growth and feed intake. A total of 199 pigs from four different sire groups (DNA Genetics Line 600) High Feed Intake/High Growth (HIHG), Low Feed Intake/High Growth (LIHG), High Feed Intake/Low Growth (HILG), and Low Feed Intake/Low Growth (LILG) were utilized at the UNL ENREC farm over 127 days. The NU*track* system allowed for individual monitoring of pigs in group housing to track daily activity traits. In total, HIHG pigs travelled less (*p* < 0.05; 139 vs. 150 km), spent more time lying (*p* < 0.05; 2421 vs. 2391 h), and less time eating (*p* < 0.05; 235 vs. 243 h) when compared to LILG pigs across time. The results suggest variation in activity occurs across the progeny of the sire groups selected to differentiate in growth and feed intake.

## 1. Introduction

The variable cost of producing market hogs is largely centered around feed cost, with diet formulation and feed conversion being the two largest contributors. Feed cost has been estimated to account for 60 to 70% of production costs, with efforts to reduce it being at the forefront of competitive systems [1]. Diet cost can be manipulated in many ways, such as utilizing alternative feed ingredients, replacing traditional protein sources with crystalline amino acids, altering feed particle size, and/or through enhanced management of animals and their environment. Altogether, the inclusion of dried distiller grains with solubles (DDGS; a major co-product from the production of ethanol from grain) is one of the most common ways to mitigate feed costs [2]. However, inclusion rates above 15% of the diet can have diminishing returns towards efficiency [3,4]. Feed conversion ratio (FCR) is commonly described by the proportion of average daily feed intake (ADFI):average daily gain (ADG) with improvements being made through genetic selection, management practices, and nutrition. An increase in FCR has traditionally been an area of focus for genetic improvement. Unfortunately, diminishing returns have been observed in response to the intense selection of FCR due to underlying correlations between growth and carcass traits [5]. For example, Godinho et al. [6] reported negative genetic correlations between FCR and ADG (−0.32) and between FCR and protein deposition (−0.47). Additionally, FCR showed positive genetic correlations between backfat depth (BF) (+0.16) and average daily feed intake (ADFI) (+0.54) [6]. Furthermore, roughly a third of the phenotypic variation associated with ADFI is accredited to residual feed intake (RFI) [7,8]), making the accurate prediction of growth performance and potential genetic selection difficult. Residual feed intake is defined as the feed consumed above expected requirements for production and maintenance [9]. Genetic selection for lower RFI has been emphasized to create offspring that consume less feed without sacrificing growth [10]. Selection for increased feed efficiency based on RFI results has been shown to improve efficiency in the grow-finish stages of production but is suggested to be combined with the growth rate in the selection scheme in order to be applicable to producers [11]. Without this integral step, producers would most likely observe decreases in growth performance due to unfavorable associations. Factors affecting RFI include digestion efficiency, metabolism, maintenance, and activity. Each of these factors has been shown to have a high level of variability among individuals [12,13]. Specifically, maintenance represents a significant portion of the daily energy intake in a pig, with roughly 34% of the daily energy intake of a 70 kg pig being directed to maintenance [14]. Typically, the activity component is disregarded due to the lack of technology to objectively measure it. To be more precise in estimating expected feed intake and to more accurately select for RFI associated traits, further understanding is needed in the area of calorie expenditure due to activity patterns.

In the past, swine production systems relied solely on human observation for monitoring behavior and activity trends within group-housed animals, with activity being a secondary concern in most circumstances [15]. Most of these observational times are dedicated to health and welfare, with caretakers focusing on behaviors that can be indicative of injury, sickness, and/or stress. It has been estimated that in modern commercial farms, human observation per animal each day can be as low as only a few seconds [16]. Even if more human interaction/labor were dedicated to observing general behavior patterns that may indicate activity levels, the presence of a human can significantly impact the adaptive behavior of the pigs [17]. Furthermore, human observation is often heavily subjective, with technicians varying in their observations. Detection of changes in behavior and activity patterns of individual pigs housed in groups is crucial to maintaining animal health and wellbeing standards. Such changes can be indicative of internal/external stressors impacting the pig. Animal-based indicators offer the most centralized measure of animal wellbeing [18]. Rapid detection of such indicators may lead to faster response time in treatment of unwell animals.

One developing solution to overcome the constraints of human observation is the progression of precision livestock technology (PLT). A primary objective with PLT is to develop a real-time/on-line system that has capabilities to maintain the individual animal ID, track traits of interest on an individual basis, and provide accurate predictions of variable change [19]. Not only will PLT allow producers to more accurately measure and assess key traits of interest, PLT can be used to improve the health and well-being of the herd. Advancements have shown that such technologies may be able to identify an animal in need of treatment prior to human observation. In the beginning, attempts to solve these PLT goals included attaching a wide array of electronic devices to individual animals, such as radio-frequency identification (RFID) collars and tags [20,21,22]. Unfortunately, these devices not only require significant maintenance and financial commitment to utilize, but also can be invasive to the animal, creating a welfare concern. Thus, recent approaches are gravitating toward non-invasive, vision-centered methods [23]. Recent developments in advanced computer vision systems, such as NU*track* (NU*track* Livestock Monitoring), now allow for activity traits to be quantified and tracked individually in a traditional commercial setting [24]. NU*track* is a fully convolution machine learning program for the long-term location and activity monitoring of individual pigs [17]. Therefore, the aim of this study was to utilize NU*track* to objectively measure nursery and finisher activity levels of pigs from sires with different estimated breeding value (EBV) combinations for growth and feed intake. The EBV is the calculation of the animal’s genetic worth based on performance records of itself and other relatives and is reported as a deviation from the population mean [25]. Additional objectives included determining variation across activity traits of all animals as well as evaluating the impact of sex on performance and activity traits.

## 2. Materials and Methods

All procedures involving the use of animals were approved by the University of Nebraska Institutional Animal Care and Use Committee protocol number 2089. A total of 25 litters [Duroc x (York x Landrace)] were utilized at the Eastern Nebraska, Extension and Education (ENREC) swine farm (May 2021). All litters were sired by Line 600 boars (Duroc terminal line boars) from DNA Genetics (Columbus, NE, USA) that had previously met selection criteria for use in commercial boar studs. A total of 287 duroc boars were available for use following selection at the boar stud. Additionally, these boars were actively being collected and used in commercial production. Sires were subdivided into one of four different sire groups based on their EBVs for average daily gain (ADG) and feed intake (FI) with the groups named as follows: High Feed Intake/High Growth (HIHG), Low Feed Intake/High Growth (LIHG), High Feed Intake/Low Growth (HILG), and Low Feed Intake/Low Growth (LILG). Each sire group was composed of pooled semen from 8 boars and boars were placed into groups based on their EBVs for feed intake and then growth. The resulting differentiation of high vs. low amongst groups shall be looked at in regard to intake firstly, and then growth secondly within intake, with the understanding that the categorization between “High” and “Low” across groups may be somewhat dissimilar. The averages of all boars available for selection for growth and feed intake EBVs were 32.2 and 16.0, respectively. The average EBV for growth and feed intake for each group can be found in Table 1.

One day post-partum, all piglets received individual identification via barcoded ear-tags. Prior to weaning, 192 pigs were selected for the trial to assess activity for both the nursery and finisher stages. Inclusion criteria included viability level, sire group, and sex. Twenty-four males and 24 females were selected per sire group. Pigs to serve as potential replacements were also identified prior to weaning, however these animals were placed in a separate nursery without the NU*track* system and thus without nursery activity data. Nursery pens (*n* = 12) in a single nursery room were stocked after the collection of individual weaning weights, with two barrows and two gilts per sire group represented per pen (16 pigs/pen; 0.42 m^2^/pig). Random allocation was utilized in placing pigs in their respective pens. Prior to entering the pen, pigs were given an individual ear tag (NU*track* tags) that was unique from pen contemporaries in either its letter/number and/or color combination [26]. Pigs were individually weighed and transferred from nursery to finisher rooms (*n* = 3 rooms, 8 pens per room) after 42 days in the nursery. Each room shared like environments. Each nursery pen was split evenly into two groups upon entry to the finisher, with one barrow and one gilt of each sire group being represented per pen (*n* = 8; 0.92 m^2^/pig). Room and pen allocation per nursery group to finisher pens was also random. To keep pens balanced for pigs/pen at the transition to the finisher, 7 pigs that were preselected as replacements at weaning were utilized to substitute for pigs that died in the nursery or were not thrifty enough to be placed into the finisher. Replacement pigs were selected to maintain the sex and sire group of the fallout pig. These animals were eventually disregarded in the compiled activity data due to their lack of nursery tracking. Pigs were off-tested after 85 days in the finisher, making the total time under analysis 127 days. At off-test, individual weights along with backfat depth (BF) and loin eye area (LEA) were recorded utilizing an ExaGo veterinary ultrasound scanner (IMV Imaging, Rochester, MN). Feed was supplied on an ad libitum basis using a standard multi-hole feeder (*n* = 8 holes in nursery and 4 holes in finisher) thus not allowing for individual feed intake records. Pen feed intake was not recorded, as pigs from all sire groups were represented in each pen evenly.

The NUtrack system was designed specifically for animals living in fixed group-housing environments via the multi-object tracking method [27]. In both the nursery and finisher rooms, each pen was equipped with a Lorex 4K Ultra HD (Lorex Technology, Inc., Markham, ON, Canada) camera mounted from the ceiling above the center of the pen, along with two infrared lights (placed on both sides of the camera) to allow for tracking to take place at night. Each camera was positioned so the field of view included the entire living space. Cameras were hardwired back to a Network Video Recorder (NVR; Lorex Technology, Inc.), which had internet capabilities to allow for the video feed to be pushed out for data processing via an ftp connection. This method of long-term tracking of individual animals within a group-housed setting is feasible via deep convolutional neural networks. These networks locate individual targets (animals) within a fixed living space and classify their identity [27]. This detection utilizes deep learning from annotated images, in which it joins body parts (left ear, right ear, shoulder, and tail) into “instances” via part association vectors [27]. Additionally, activity tracking is processed via the Bayesian multi-object tracking method. Utilizing frame-to-frame movement probabilities with images being captured at 5 frames per second, the probabilities of individual identification (by the NU*track* ear tags) can be assessed. Across a variety of environments, this method has shown to be over 90% accurate in maintaining individual identification [27]. In short, NU*track* does not classify pig behavior, instead it tracks activity and body posture determined by body part position and change in location based on frame-to-frame movement analysis. Currently, activities tracked on an individual pig basis with NU*track* include standing time, sitting time, lying time (sternal and lateral), time spent engaged at the feeder, velocity when walking, rotations, and distance traveled. Time spent engaged at the feeder tracks the amount of time an individual’s head remains positioned appropriately for eating within the designated area surrounding the feeder. Activity traits used for analysis included standing time, lying time, and distance travelled. All other traits not further discussed in this manuscript were not significantly associated with traits included in this study’s objectives. Variables were analyzed on a per day basis, with the periods considered being those of nursery, finisher, and cumulative. Cumulative totals included individual pig as the unit of replication. Pigs in this analysis were of the same genetics and housed in the same rooms/pens with the same lighting and stocking density in which the NU*track* system was originally trained on. Furthermore, NUtrack has been shown to show greater specificity and sensitivity than trained human observers while providing continuous coverage and analysis of pig activity [28] that is not feasible for human observers to accomplish in a study of this scale.

RStudio v. 1.1.456 was used for data editing and analysis [29]. Together, the lm() and emmeans() functions were used to derive estimated marginal means. Analysis of variance (ANOVA) testing was conducted utilizing individual pig data as the unit of measure with the use of the summary() function. Pigs that suffered mortality, lameness, and/or ruptures were removed from the data set. A total of 175 pigs were available for analysis after editing. A decrease in observations was the result of mortality, removal from pen for health/lameness, and non-sensical outliers such as when it was confirmed that a pig had died the day before it was noted by farm staff, as the death occurred after they had exited out for the day. Video data was disregarded for the day of entry to nursery as well as the day of transition from nursey to finisher due to incomplete 24-h time frames in their respective pen; although future investigation into these time periods may prove to be informative. Categorical fixed effects for activity traits included sire group, sex, and location (room and pen). Average daily gain was calculated as the difference in off-test weight from weaning weight divided by 127 days. The analysis for nursery growth included the categorical fixed effects for sire group and sex with the covariate of weaning weight. The analysis for finisher growth included the categorical fixed effects for sire group and sex with the covariate of nursery exit weight. For carcass traits, the analysis included the categorical fixed effects of sire group and sex with the covariate of off-test weight. No carcass data was available for the investigation into carcass quality given the activity levels and sire group.

## 3. Results

During the nursery phase, HIHG pigs travelled less (*p* < 0.05) and spent more time lying per day (*p* < 0.05) compared to the other three groups. HIHG pigs spent more time at the feeder per day compared to HILG and LIHG pigs (*p* < 0.05) but this was less (*p* < 0.05) than that observed in both HILG and LIHG pigs. Furthermore, HILG pigs travelled more per day (*p* < 0.05) compared to all other groups. The estimated marginal means activity traits in the nursery can be found in Table 2.

Throughout the finisher stage, HIHG pigs travelled less per day (*p* < 0.05) compared to all groups and spent more time lying per day (*p* < 0.05) and spent less time engaged at the feeder (*p* < 0.05) compared to the LIHG and LILG groups. Conversely, LILG pigs travelled the most (*p* < 0.05), spent the least amount of time lying down (*p* < 0.05), and spent the most time engaged at the feeder (*p* < 0.05) out of all groups. Estimated marginal means activity traits in the finisher can be found in Table 3.

In total, when combining nursery and finisher stages, HIHG pigs travelled less (*p* < 0.05) and spent more time lying (*p* < 0.05) than the LIHG and LILG groups as well as spent less time engaged at the feeder (*p* < 0.05), compared to all other groups. Contrarily, LILG pigs travelled more (*p* < 0.05) than all other groups throughout both stages. The estimated marginal means for cumulative totals for both stages combined can be found in Table 4.

When looking at the performance data across groups, total ADG was the highest (*p* < 0.05) in HIHG pigs and the lowest (*p* < 0.05) in LILG pigs; whereas HILG and LIHG pigs were similar. HIHG pigs had the smallest (*p* < 0.05) LEA, whereas LILG pigs had the largest (*p* < 0.05) LEA; and HILG and LIHG pigs were intermediate in comparison. LILG pigs had the least (*p* < 0.05) amount of backfat, whereas LIHG pigs had the most (*p* < 0.05) backfat and HIHG and HILG were the same. The results showing the ADG, LEA, and BF estimated marginal means can be found in Table 5.

Activity and performance averages between the sexes were also calculated. Barrows had a greater (*p* < 0.01) total ADG and more (*p* < 0.01) BF at off-test, yet gilts had a larger (*p* < 0.01) LEA, as expected. In terms of activity, gilts travelled more (*p* < 0.01) throughout the day and spent less (*p* < 0.01) time engaged at the feeder compared to barrows. No significant differences were seen between the sexes for time spent lying per day. The estimated marginal means for performance and activity traits for each sex can be found in Table 6.

## 4. Discussion

In the current study, we utilized an advanced computer vision system (NU*track*) to detect activity differences in pigs from sires with different EBV combinations for growth and feed intake in the hopes of understanding if these EBVs are predictive of activity differences. The sire groups were composed of boars with contrasting EBVs for both growth and feed intake; two traits commonly measured and selected upon to increase performance of terminal animals. These results show clear differences in activity between the most extreme groups for both feed intake and growth (HIHG vs. LILG) but showed inconsistencies when comparing the intermediate groups (HILG vs. LIHG). A more comprehensive comparison among these groups could be made by including a larger sample size, as the EBV difference achievable in this population between these groups for ADG was quite small. Alternatively, an unselected population with greater genetic variation could be used to form EBV groups. The sires used for this study were boars that had previously met selection criteria for use as an AI sire; thus, the variation of performance traits was decreased. Furthermore, some overlap was present in the categorization of sire groups for growth and feed intake, potentially hindering the resulting differences amongst them. Together, further work is needed to properly delineate the association between individual EBVs and the corresponding activity trait(s).

Pigs sired by HIHG boars spent more time lying, travel less, and spend less time engaged at the feeder on average compared to pigs from most other sire groups. Differences across sire groups were largely consistent between the nursery to the finisher stage, with some variation explained by the larger number of days and the changing rate of growth and feed intake. In total, HIHG pigs walked 10.92 km less over both phases, as well as spent 29.63 more hours lying down and 8.46 less hours engaged at the feeder compared to LILG pigs. It may appear counter intuitive that HIHG pigs spend less time at the feeder than other groups; however, it has been demonstrated that pigs with a higher intake per second of feed are faster growing and have a greater deposition of adipose tissue [30]. These differences, especially the difference in distance traveled and time spent lying down, can be assumed to impact calorie expenditure and may explain a portion of the variation observed in performance. the non-independence amongst activity traits such as distance travelled and lying time should also be noted. If an animal spends more time lying down, there will be less time available for compiling movement. This time allowance for activity may explain some of the variation. Differences observed in time spent at the feeder across sire groups may be due to biological effects. Literature has shown that pigs will spend up to half of their active time on “foraging” and eating, with significant variability amongst individuals [31]. The rate at which individuals were eating is most likely the largest factor contributing to the variation observed within and across groups. Additionally, the frequency in which pigs consumed feed may impact carcass and other performance traits [32]. Activity differences were less clear between HILG and LIHG groups. Differences between these two groups were often small/non-significant. Perhaps the smaller differences in mean EBVs, most notably in growth, can explain some of the similarities in the activity patterns. The variability present across and between sire groups may represent an opportunity for genetic selection of an individual trait or an index of activity traits in the future. As individual feed intake data was not collected and pigs of all groups were equally represented in each pen, we could not determine differences in feed efficiency among these groups nor individuals within groups.

The sample size of the study is small relative to traditional studies involving EBVs, yet the difference in growth between the HIHG (0.88 kg/d) and LILG (0.81 kg/d) groups of 0.07 kg/d was half of the difference in the intake EBVs across respective sire groups. This is what was predicted from the difference in EBV for growth, as half of the progeny’s breeding value for any given trait is assumed to come from the sire and half from the dam. HIHG pigs also had more (*p* < 0.05) backfat at off-test (1.48 vs. 1.42 cm), which suggests that the activity levels of the pigs could partially explain fat and lean deposition in growing pigs. A larger deposition of carcass fat can be explained by the rate of fat synthesis being greater than the rate of fat utilization [33]. Significant differences were also observed for LEA, with LILG pigs (52.7 cm^2^) being greater (*p* < 0.05) than HIHG pigs (49.66 cm^2^). This agrees with previous work, as physical activity has been shown to positively impact the leanness of carcass composition [34]. No difference was seen in ADG between HILG- and LIHG-sired pigs which again may be attributed to the small difference in the growth EBV.

The comparison of performance between barrows and gilts coincides with what is commonly seen in the industry. The results here show that barrows have greater ADG, more backfat, and a smaller LEA on average compared to gilts. This agrees with the review done by Kansas State University [35] of 34 different trials showing gilts having a 5.9% lower ADG, 11.7% less backfat and a 4.5% increase in lean percentage compared to barrows. However, activity differences associated between sexes have yet to be quantified or managed. These results show that gilts are more active (50.9 m/d) and spend less time at the feeder (−7.1 min/day) compared to barrows over both phases of production. No meaningful difference was detected between lying time per day (2 min/d). As mentioned earlier, individual feed intake data is needed to accurately estimate feed efficiency between sexes. However, these results show variability among the sexes, again representing opportunity for selection in the future and the potential to create different management strategies such as providing more feeder space per pig for barrows versus gilts given their increased time engaged at the feeder. Overall, a more expansive study is needed to better understand the significance of all associations mentioned above.

## 5. Conclusions

Calorie expenditure due to activity in market-driven animals has been difficult to quantify in the past due to the absence of technology to objectively measure it in a commercial setting, but it is understood to impact feed efficiency. The results herein suggest that the progeny activity level is impacted by the sire EBVs for production traits and could be detected in the progeny of boars already selected for use in commercial production where the genetic variation is less than the full population’s genetic variation. In the current study, pigs from boars with superior growth and feed intake EBVs were less active and faster growing compared to their contemporaries. In addition, pigs sired by boars with the lowest EBVs for growth and feed intake were the most active and slowest growing. Further work may lead to genetic selection for activity traits to decrease RFI levels and ultimately improve production rates through reduced caloric expenditure. In addition, the results show differences amongst sexes in terms of activity and growth, with barrows being less active and faster growing when compared to gilts. Further work in the area of activity and behavior differences amongst sexes may result in different management strategies for feeding and housing of different sexes. Ultimately, furthering the objective tracking of activity patterns with precision livestock farming systems, such as NU*track,* may help producers increase the rate of genetic gain for feed efficiency and contribute to improved pig management.

## Figures and Tables

**Table 1 animals-13-01581-t001:** Average EBVs for growth and feed intake of each sire group.

Trait	HIHG	HILG	LIHG	LILG
Feed Intake (g/d) ^1^	197.6 ± 55.2	170.7 ± 32.0	−136.0 ± 40.6	−188.0 ± 37.3
Growth (g/d) ^1^	101.8 ± 8.4	39.5 ± 10.6	27.0 ± 8.4	−33.2 ± 17.0

^1^ EBVs are displayed as the difference from the population mean ± standard deviations.

**Table 2 animals-13-01581-t002:** Estimated marginal means for activity traits in the nursery stage.

	Nursery ^1^					
Sire Group	DIST (m/d)	SE	LIE (min/d)	SE	EAT (min/d)	SE
HIHG	1409 ^c^	8.9	1075 ^a^	2.6	143 ^b^	1.1
HILG	1474 ^a^	8.9	1065 ^b,c^	2.6	154 ^a^	1.1
LIHG	1442 ^b^	8.9	1063 ^b,c^	2.6	158 ^a^	1.1
LILG	1430 ^b,c^	8.9	1070 ^a,b^	2.6	140 ^b^	1.1

^1^ Data reported on per day basis. Abbreviations: DIST = distance travelled; LIE = Lying time (sternal and lateral); EAT = time engaged at feeder. ^a,b,c^ Significantly different (*p* < 0.05) within trait.

**Table 3 animals-13-01581-t003:** Estimated marginal means for activity traits in the finisher stage.

	Finisher ^1^					
Sire Group	DIST (m/d)	SE	LIE (min/d)	SE	EAT (min/d)	SE
HIHG	936 ^c^	6.9	1199 ^a^	1.2	95 ^c^	0.6
HILG	985 ^b^	6.8	1197 ^a^	1.2	97 ^c^	0.5
LIHG	1002 ^b^	6.9	1187 ^b^	1.2	100 ^b^	0.6
LILG	1071 ^a^	6.7	1174 ^c^	1.2	104 ^a^	0.5

^1^ Data reported on per day basis. Abbreviations: DIST = distance travelled; LIE = Lying time (sternal and lateral); EAT = time engaged at feeder. ^a,b,c^ Significantly different (*p* < 0.05) within trait.

**Table 4 animals-13-01581-t004:** Cumulative totals for estimated marginal means of activity traits across both nursery and finisher stages.

	Cumulative Nursery & Finisher ^1^					
Sire Group	DIST (km)	SE	LIE (h)	SE	EAT (h)	SE
HIHG	139 ^c^	0.78	2421 ^a^	4.2	235 ^c^	1.3
HILG	145 ^b,c^	0.79	2417 ^a^	4.2	245 ^b^	1.3
LIHG	145 ^b^	0.79	2398 ^b^	4.2	252 ^a^	1.3
LILG	150 ^a^	0.78	2391 ^b^	4.1	243 ^b^	1.3

^1^ Data reported as a cumulative total across both production phases. Abbreviations: DIST = distance travelled; LIE = Lying time (sternal and lateral); EAT = time engaged at feeder. ^a,b,c^ Significantly different (*p* < 0.05) within trait.

**Table 5 animals-13-01581-t005:** Estimated marginal means of performance traits for each sire group.

	Performance Traits					
Sire Group	ADG (kg/d)	SE	LEA (cm^2^)	SE	BF (cm)	SE
HIHG	0.88 ^a^	<0.01	49.66 ^d^	0.07	1.48 ^b^	<0.01
HILG	0.85 ^b^	<0.01	50.28 ^c^	0.07	1.48 ^b^	<0.01
LIHG	0.85 ^b^	<0.01	51.24 ^b^	0.07	1.52 ^a^	<0.01
LILG	0.81 ^c^	<0.01	52.78 ^a^	0.07	1.42 ^c^	<0.01

Abbreviations: DIST = distance travelled; LIE = Lying time (sternal and lateral); EAT = time engaged at feeder. ^a,b,c,d^ Significantly different (*p* < 0.05) within trait.

**Table 6 animals-13-01581-t006:** Estimated marginal means for performance and activity traits of each sex.

Trait	Barrows	SE	Gilts	SE	*p<*
ADG (kg/d)	0.88	<0.01	0.82	<0.01	<0.01
BF (cm)	1.55	<0.01	1.39	<0.01	<0.01
LEA (cm^2^)	50.60	0.05	51.37	0.05	<0.01
DIST (m/d)	1127.31	4.35	1178.29	4.1	<0.01
LIE (min/d)	1148.65	1.02	1146.21	1.01	0.39
EAT (min/d)	120.91	0.46	113.33	0.45	<0.01

Abbreviations: ADG = nursery and finisher average daily gain (kg/day); LEA = Loin Eye Area (cm^2^) at off-test, BF = backfat (cm) at off-test; DIST = distance travelled per day (meters); LIE = Lying time (sternal and lateral) per day (minutes); EAT = time engaged at feeder per day (minutes).

## Data Availability

Data available upon request.

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
