# Peer review of "Utilizing NU*track* to Access the Activity Levels in Pigs with Varying Degrees of Genetic Potential for Growth and Feed Intake"

_animals, 2023, doi:10.3390/ani13101581_

Round 1

Reviewer 1 Report

The manuscript evaluates the association between activity level in pigs and breeding values of sires for feed intake and growth rate. Activity is measured by image analysis using individual animal tracking in the offspring of sires with extreme EBV values. The manuscript is clear and well-written and provides interesting results, however, some aspects of the paper should be improved.

Major comments:

L110-116: How do the EBV values in the four experimental groups of sires compare to the average and range of the breeding candidates? What’s the rank of the sires in the most extreme groups?

Table 1: please add also a measure of the variability in the EBV for the 4 groups.

L165-169: the response variables listed here do not correspond to the variables investigated, please clarify why some variables were excluded from the analysis. In addition, please add more details on the criteria used to classify pig behavior. Also, clarify that the variables analyzed referred both to the average daily time or distance and total time and distance, and that the periods considered in the analysis were nursery, finisher and cumulative.

L170-182: what was the method used? Was it a GLM? How was the post hoc testing performed? Please add the name of the R package used for the statistical modeling and post hoc testing.  

L170-182: what were the criteria used for outlier elimination? How many outliers were removed?

L177: was does the location refer to? Does it correspond to the pen?

L179-181: the authors indicate that weaning weight, nursery exit weight, and off-test weight were included in the model as categorical effects. If so, please include the number and interval of the weight of the categories.

Results: the results obtained for some of the effects included in the models (location, body weight) are not discussed in the text. Please comment on this. Also, please indicate the R2 of the models.

Table 2: please add the standard errors of the estimates and consider transforming the table into a figure for more immediate readability of the results.

L201-211: this seems to be a little “unrelated” to the rest of the paper and results merely depend on the genetic correlation between feed intake/growth and ADG/LEA/BF. Is this relevant to the aim of the manuscript?

L224-241: the contents of this paragraph are already reported in the introduction and should not be repeated in the discussion. Please remove.

L246-248: Was the difference between EBV in the four groups significant? This information could help the discussion of the results.

L258-260: how do the authors explain that high-intake pigs were those spending less time at the feeder? Does this agree with the literature on pig feeding behavior?

Minor comments:

Abstract: the abstract does not mention results but only materials/methods and conclusions. I suggest reporting the difference between groups along with the corresponding p-value.

L104-105: please add more information about the DNA Genetics Line 600 (e.g. terminal line, producer, location of the nucleus farm)

L244: I suggest replacing “exterior” with “most extreme groups for both intake and growth”.

Author Response

The authors would like to thank the reviewer for the thorough and thoughtful review that will enhance the manuscript's value and readability. Our detailed response follows:

L110-116: How do the EBV values in the four experimental groups of sires compare to the average and range of the breeding candidates? What’s the rank of the sires in the most extreme groups?

  • While we understand the desire to know where the boars rank for the individual components, the boars (n=32) were selected out of 287 boars to provide the most divergent groups possible of the combination of both feed intake and average daily gain from a population of boars already selected for use in production and therefore deemed of sufficient quality to be used in commercial production. Rankings for the boars is not as meaningful to expected progeny differences as the EBV’s and the now provided SD of those groups per the reviewers suggestion which should provide the differences in performance the reviewer was desiring. We also provided the EBV means for both growth and feed intake of the boar population in lines 130-131 to help demonstrate the diversity of the group.

Table 1: please add also a measure of the variability in the EBV for the 4 groups.

  • We included standard deviations in table 1 for EBV’s of the 4 groups.

L165-169: the response variables listed here do not correspond to the variables investigated, please clarify why some variables were excluded from the analysis. In addition, please add more details on the criteria used to classify pig behavior. Also, clarify that the variables analyzed referred both to the average daily time or distance and total time and distance, and that the periods considered in the analysis were nursery, finisher and cumulative.

  • We reworded the manuscript to clarify that we are not classifying behavior per se but activity/body posture as determined by their body part position and change in location based on frame by frame movement analysis.
  • We noted that only significant activity traits pertaining to the study were reported with non-significant traits omitted.
  • We added information as to how the system identifies individual pigs body posture so that the reader would not need to reference the original papers on the system.
  • We clarified the average daily time/distance analyzed for each time period in the material and methods

L170-182: what was the method used? Was it a GLM? How was the post hoc testing performed? Please add the name of the R package used for the statistical modeling and post hoc testing.  

  • We included procedures and packages in final paragraph of materials and methods

L170-182: what were the criteria used for outlier elimination? How many outliers were removed?

  • Outliers were addressed in final paragraph of materials and methods and clarified that outlier data proved to be where actual time of death happened between the time when workers left the facility in the afternoon and when they entered the facility the next work day, ie the time of death was actually the day before the farm staff noted it.

L177: was does the location refer to? Does it correspond to the pen?

  • Location was clarified in final paragraph of material and methods section to be pen (in the nursery) and room and pen (in the finisher).

L179-181: the authors indicate that weaning weight, nursery exit weight, and off-test weight were included in the model as categorical effects. If so, please include the number and interval of the weight of the categories.

  • Reworded to note weights were included in the model as covariates and not categorical traits we incorrectly submitted in the original manuscript.

Results: the results obtained for some of the effects included in the models (location, body weight) are not discussed in the text. Please comment on this. Also, please indicate the R2 of the models.

  • Location (pen) was not significantly associated with any trait for nursery data and therefore was not shown. Pen did have a significant effect on performance data from the finisher but as it is beyond the scope of the activity monitoring project, we did not believe it to be valuable to the readers or scientific literature and therefore chose to not report those values.
  • Body weight was included in the analysis as a covariate not as a catagorical trait as our manuscript initially suggested.
  • The R2 values of the models are admittedly small in part due to the size of the dataset and would distract from the intent of the manuscript as a whole. Further work in progress by the authors on a larger unselected population of pigs will provide the scientific community better estimates of R2 in swine.

Table 2: please add the standard errors of the estimates and consider transforming the table into a figure for more immediate readability of the results.

  • Added standard errors to all the tables. Given the scale of the data, the authors believe figures would not enhance reader experience. We therefore split the table into additional tables by phase of production in an effort to address the reviewers concern.

L201-211: this seems to be a little “unrelated” to the rest of the paper and results merely depend on the genetic correlation between feed intake/growth and ADG/LEA/BF. Is this relevant to the aim of the manuscript?

  • Producers value the inclusion of ADG/LEA/BF into manuscripts that relate to feed intake, feed conversion, and growth as both genetic and management effects can play a role in the outcomes. Additionally, the inclusion of the data provides evidence that the progeny of the sire groups followed the traditional expectations of animals of the differing intake and growth categories. Therefore, we feel there is value in this section remaining in the manuscript.

L224-241: the contents of this paragraph are already reported in the introduction and should not be repeated in the discussion. Please remove.

  • We removed the repetitive section to be more concise.

L246-248: Was the difference between EBV in the four groups significant? This information could help the discussion of the results.

  • The inclusion of the SD for the EBVs demonstrated where the sire groups were significant from each other noting that not all groups were significantly different from each other for both growth and intake. Selecting boars that truly fell into the four quadrants of the full genetic population was not possible as the boars available for use were already preselected to be superior enough to be used in commercial production. A significant difference was not expected between all traits for all sire groups. We sought to note this expectation better in this iteration of the manuscript.

L258-260: how do the authors explain that high-intake pigs were those spending less time at the feeder? Does this agree with the literature on pig feeding behavior?

  • We addressed this second paragraph of the discussion and added references showing where it agrees with literature.

Abstract: the abstract does not mention results but only materials/methods and conclusions. I suggest reporting the difference between groups along with the corresponding p-value.

  • We added in results to the abstract.

L104-105: please add more information about the DNA Genetics Line 600 (e.g. terminal line, producer, location of the nucleus farm)

  • We added information on the boar line, company, and company location in the material and methods.

L244: I suggest replacing “exterior” with “most extreme groups for both intake and growth”. 

  • We made the change as suggested.

Reviewer 2 Report

Monitoring activity levels in pigs by sex and with varying EBV's for growth and feed intake using NUtrack 

The article titled "Monitoring activity levels in pigs by sex and with varying EBV's for growth and feed intake using NUtrack" describes the use of an advanced computer vision system, NUtrack, to monitor activity levels in pigs across different sex and sire groups with varying expected breeding value combinations for growth and feed intake. The study utilised 199 pigs from four different sire groups and evaluated activity levels through frame-to-frame movement probabilities, with images captured at 5 frames per second, to maintain individual identification of the pigs through the NUtrack ear tags. 

The authors suggested that NUtrack accurately tracks various pig activities, such as standing, sitting, lying, feeding, walking, rotations, and distance travelled. However, my concerns are mainly about the study's use of NUtrack without rigorously scrutinising its performance to validate its effectiveness on their dataset, which could raise doubts about the validity of the study's results. 

Further to the above and from the computer vision novelty aspect of the work, I also noted that the study did not build on the work of a previous study by Psota et al. (2020) that developed keypoint detection and MAP estimation for individual animal identification.

Author Response

The authors suggested that NUtrack accurately tracks various pig activities, such as standing, sitting, lying, feeding, walking, rotations, and distance travelled. However, my concerns are mainly about the study's use of NUtrack without rigorously scrutinising its performance to validate its effectiveness on their dataset, which could raise doubts about the validity of the study's results. 

  • We made a note (materials methods section, paragraph that pertains to NUtrack) regarding the pigs used under analysis being of the same genetics and housed in the same room/pen that the system was trained on. Manuscripts on the technology and validation of the system were referenced within the manuscript for review. The technology used in this study provides data on every second of the time period being analyzed, something in which human annotation cannon feasibly accomplish on data of this size and duration.

Further to the above and from the computer vision novelty aspect of the work, I also noted that the study did not build on the work of a previous study by Psota et al. (2020) that developed keypoint detection and MAP estimation for individual animal identification.

  • The reviewer is correct in that this paper did not report major advancements in the computer algorithm used for the manuscript from what Psota et al (to which Mote and Schmidt were also authors on) and noting advancements of the system was not the intention of this manuscript. The intention of this manuscript was to utilize the computer algorithm in its current stage of advancement to investigate the differences in traits that NUtrack does provide data on from animals, in a production setting, to answer whether or not there are genuine differences in activity of pigs from sires genetically differing in feed intake and growth characteristics.

Reviewer 3 Report

Potentially interesting paper that needs some revision to make it comprehensible to readers of Animals

Author Response

The authors would like to thank the reviewer for a thorough and thoughful review and for offering suggestions that will enhance the manuscript for both content and readability.

This paper describes a potentially interesting application of precision farming technology (NUtrack) to the analysis of pig behaviour., which could make a significant contribution to pig breeding programs. However, I could not recommend acceptance of the paper in its present form because there is not enough information about the statistical analysis that was used on the data to be able to judge whether the reported results are valid or not.

        * The materials and methods section was substantially upgraded to address this comment to include models that were inadvertently omitted from the initial version of the manuscript.

Title. If this paper were submitted to a specialist pig journal it might be acceptable to include the acronym EBV in the title and then not to explain what this means in the text as presumably pig specialists already know what it means. However, as Animals is a general journal with a wide readership, many people will not be familiar with it. At the very least EBV should be explained when it is first introduced. The authors already use a large number of acronyms (which are explained but quite difficult to remember) but it is essential they explain all of them. I would suggest altering the title so that it is more comprehensible to readers of Animals. 

      * Title has been altered to "Utilizing NUtrack to access the activity levels in pigs with varying degrees of genetic potential for growth and feed intake" to avoid genetic jargon that non geneticists might not be familiar with.

Line 96. Please explain EBV. 

          * definition was included.

Line 106. As I understand it, the initial selection of the four groups was based on the characteristics of the sire, but these characteristics are not explained very clearly. What constitutes a sire with High Feed Intake/ Low growth rate etc? (i.e. how high or how low?).

     * The reviewer is correct in that the selection of the sires used to generate the progeny for this study were based on the genetic potential of the sires as demonstrated by the estimated breeding values. Table 1 now includes the standard deviation of the breeding values to more clearly demonstrate the genetic potential and variation of the four sire groups.

Table 1 refers to the characteristics of the offspring, not the sires.

     *table 1 is the genetic breeding value (potential) of the sires and is twice (per definition of estimated breeding value) as to what the progeny are supposed to exhibit.

How many individuals contributed to the averages?

     *We drew more attention to the sire groups being composed of 8 boars per group out of a total useable population of 287 boars available. The EBV averages for each group is listed in table 1.

What was the ‘population’ with which they were compared?

     *We included a line in the materials and methods to note there was 287 Duroc boars available to be selected from.

How much variation (in the deviations) was there? This could be important because it could make a difference to the interpretation of the results whether there was a lot of overlap or none at all.

     * The standard deviation for both growth and intake EBVs were listed for each sire group in table 1 and does indicate some overlap between the groups for certain traits.  The discussion section also now draws the readers attention to that fact and why not all groups exhibit significantly different production levels.

Line 183. What statistical tests were done to achieve these results? All that is given here is a P value, but this does not mean anything without knowing what was done! It is good practice to give the result, the test used, the degrees of freedom and then the p value. Without this basic information it is impossible to evaluate the results.

     *This was a major oversight on our part on the initial manuscript. We included information on ANOVA testing and models in materials and methods.

Were pens or individual pigs taken as the unit of replication? Did each pig contribute one cumulative total or did each pen contribute one cumulative total (n= 6/group)?

     * We clarified that individual pigs were the unit of replication for the study in the materials and methods section.

Discussion
This should include some discussion of the non-independence of time budget data. If an animals spends a lot of time doing A, it has less time available for doing B so A and B are not independent.

     *We included discussion on the principle of non-independence of the time budget on the data and that the time of our activity traits on not completely independent from each other.

Round 2

Reviewer 2 Report

I appreciate your efforts to revise the manuscript accordingly. I acknowledge the note added in the materials methods section regarding the genetics and housing conditions of the pigs used in the analysis (but what about age and lighting environment?). I would like to emphasise the importance of validating the performance and precision of NUtrack on your dataset to ensure the reliability of the study's results. Including information on the margin of error and the validation of the algorithm's effectiveness would add robustness to your findings.

Author Response

I appreciate your efforts to revise the manuscript accordingly. I acknowledge the note added in the materials methods section regarding the genetics and housing conditions of the pigs used in the analysis (but what about age and lighting environment?). I would like to emphasise the importance of validating the performance and precision of NUtrack on your dataset to ensure the reliability of the study's results. Including information on the margin of error and the validation of the algorithm's effectiveness would add robustness to your findings.

We added further information in the materials and methods to further note the similarities of the pigs and environment the system was originally trained on to include lighting and stocking densities. We also referenced a manuscript [28] that also utilized NUtrack in this same environment that compared trained human observers with NUtrack and noted NUtrack provided greater sensitivity and specificity than trained human observers as well as can provide 24/7 coverage of large data sets over an extended time frame where humans simply do not have the time/resources to fully offer that level of annotation. We appreciate the reviewers concern for validation of the algorithm used in this manuscript which hopefully the additional reference will suffice but note that it is impractical to provide the level of scrutiny the reviewer desires for each set of pigs tracked with the computer vision program already trained for the environment.

Reviewer 3 Report

Still a lot of abbreviations to remember for someone unfamiliar with them!

Authors have addressed most of the issues raised. The presentation of statistical results is not very clear. I have asked the editors whether Animals has a standard way of presenting results. I would expect this from a journal. If they do, the authors should adopt this.

Author Response

Still a lot of abbreviations to remember for someone unfamiliar with them!

- We ensured that all abbreviations were spelled out at least the first time they were utilized. Widely accepted trait abbreviations were utilized where appropriate. If the editors so choose, we can exchange the abbreviations utilized for NUtrack traits to their full name. We could have utilized generic group numbers/letters for the sire groups but felt that utilizing their breeding values best described the groups as to where readers would not have to reference the materials and methods each time to remember what each group was composed of.

Authors have addressed most of the issues raised. The presentation of statistical results is not very clear. I have asked the editors whether Animals has a standard way of presenting results. I would expect this from a journal. If they do, the authors should adopt this.

- If the editors prefer a different format for presenting results, we will adapt.